# Evaluation of Ecological Quality Status and Changing Trend in Arid Land Based on the Remote Sensing Ecological Index: A Case Study in Xinjiang, China

Yimuranzi Aizizi [1], Alimujiang Kasimu [1,2,3,*], Hongwu Liang [4], Xueling Zhang [1], Bohao Wei [1], Yongyu Zhao [4] and Maidina Ainiwaer [1]

1   School of Geography and Tourism, Xinjiang Normal University, Urumqi 830054, China;
    amirhamza@stu.xjnu.edu.cn (Y.A.); zhangxueling@stu.xjnu.edu.cn (X.Z.); weibohao@stu.xjnu.edu.cn (B.W.);
    medinaanwar@stu.xjnu.edu.cn (M.A.)
2   Research Centre for Urban Development of Silk Road Economic Belt, Xinjiang Normal University,
    Urumqi 830054, China
3   Xinjiang Key Laboratory of Lake Environment and Resources in Arid Zone, Urumqi 830054, China
4   State Key Laboratory of Desert and Oasis Ecology, Key Laboratory of Ecological Safety and Sustainable
    Development in Arid Lands, Xinjiang Institute of Ecology and Geography, Chinese Academy of Sciences,
    Urumqi 830011, China; lianghongwu23@mails.ucas.ac.cn (H.L.); zhaoyongyu23@mails.ucas.ac.cn (Y.Z.)
*   Correspondence: alimkasim@xjnu.edu.cn; Tel.: +86-150-9907-9312

**Abstract:** Ecosystems in arid areas are under pressure from human activities and the natural environment. Long-term monitoring and evaluation of arid ecosystems are essential for achieving the goal of sustainable development. The Xinjiang Uygur Autonomous Region (Xinjiang) is a typical arid region located in Northwest China with a relatively sensitive ecosystem. Under the support of the Google Earth Engine (GEE) cloud platform's massive data collection, the remote sensing ecological index (RSEI) from 2000 to 2020, both in summer and spring, is established, and the variation trend of the ecological quality in Xinjiang is evaluated by coefficient of variation (CV), Sen's slope analysis, Mann–Kendall trend test (M–K test) and Hurst index. In addition, a partial correlation analysis is processed between RSEI and selected climatic factors, including precipitation and temperature, to find out the mode of correlation between ecological quality and the natural climate. In the last two decades the following has become apparent: (1) The RSEI values of Xinjiang have been relatively low and unstable both in summer and spring, with a trend toward increasing; (2) The distribution characteristics of RSEI levels both in summer and spring have been similar; low levels were concentrated in the desert and wilderness, while high levels were concentrated around the oasis; (3) The ecological quality in Xinjiang has been relatively stable, with a trend of sustained increase both in summer and spring. There was also a small area of sustained decrease around the Junggar Basin and Turpan Basin in summer and a small area of significant decrease in the center of the Taklamakan Desert in spring; (4) In summer, the precipitation has obviously positively correlated in the Southwest. The temperature has obviously positively correlated in the northwestern part; in spring, the precipitation has obviously positively correlated in the Western part; the temperature has obviously positively correlated in the oasis around the Yili River Basin and Tarim Basin.

**Keywords:** change trend analysis; ecological quality; partial correlation analysis; RSEI; Xinjiang Uygur Autonomous Region

## 1. Introduction

The continuous deterioration of ecological quality caused by the high intensity of human activities has brought obvious damage and threats to the stability of the ecological system [1,2] and has led the relationship between human beings and the ecological environment into a more sensitive and severe situation [3–5]. Meanwhile, the extreme

weather and land desertification caused by climate changes have strengthened the sense of human beings toward the crisis of ecological environment deterioration [6,7], and humans have realized that strengthening the stability of the ecological environment was a key issue that needed to be solved urgently [8,9]. Protecting the ecological environment's stability and maintaining the harmonious relationship between human beings and the ecological environment are essential elements for the sustainable development of human society in the future [10]. In this regard, to improve the priority of ecological environment protection, the Chinese government has proposed the development guideline of giving high priority to ecological protection and actively practicing the sustainable development concept of "both gold and silver mountains, but also lucid water and lush mountains" [11–13].

The scientific assessment of regional ecological environment quality plays a crucial role in the formulation and implementation of ecosystem protection planning [14,15]. Therefore, accurately understanding the overall status of ecological environment quality and monitoring the trend of ecological environment changes have a significant impact on strengthening the capability of ecological environment management and finding solutions to address the key issues caused by global changes related to green development [16]. For the above concerns, how to quickly and efficiently characterize and measure the regional ecological environment quality on a spatial and temporal scale is becoming a key part of the normalized monitoring of ecological quality change trends [17]. The systematic assessment of ecological environmental quality is regarded as an urgent need to recover the damage brought by human beings to the ecological environment while maintaining the recent high level of economic development [18,19]. As an active effort in scientific response and solution to the concerns mentioned above, the government of China enacted the "Technical Specifications for Evaluation of Ecological Environment Conditions" as early as 2012, which is dedicated to successfully uniformizing the criteria of internal ecological environmental evaluation progress with scientific evaluation standards and achieving the goal of timely analysis of the ecological environmental quality in the whole country [20]. However, although the standard takes multiple critical factors into consideration, unfortunately, because of the big disparity of the data selected for the construction of the Ecological Index (EI), the inconvenience of acquiring the related data makes the process extraordinarily difficult; ultimately, the results of the EI merely acquire the numerical results once a year, making it impossible to finish the visualized analysis of the status of the regional ecological environmental quality [21,22].

Along with the quick development process of remote sensing theory and sensor technologies, the special merit of timely, macroscopic, and repeated observations of remote sensing techniques provides great convenience and a sufficient data source for ecological environment observation and evaluation. To successfully visualize the ecological quality status, some previous studies have applied one or two related factors, such as the land surface temperature (LST) [23,24], vegetation condition index (VCI), and fractional vegetation cover (FVC), to evaluate the partial ecological quality in a more efficient way [22,25,26]. However, considering the instinctive complexity and diversity of the related components of the ecological system, the importance of FVC, VCI, and LST is essential to the stability of the ecological environment, and the obvious impact of these indices on the ecological environment has been proven by multiple studies. However, these types of evaluation methods are obviously limited by the singleness of the selected factors, making it hard to support the overall evaluation of the quality of the ecological environment [27,28]. For this reason, it is requisite to establish a highly extensive and scientifically supported evaluation scheme that takes more impactful components into consideration for a detailed analysis of the quality of the ecological environment [29]. As a comprehensive assessment method of the ecological environmental quality, the Remote Sensing Ecological Index (RSEI) successfully visualizes the results of ecological quality assessment by taking the four considerable components that can be directly detected by human beings into consideration, namely, greenness, wetness, heat, and dryness [30], finally realizing the goal of a macroscopically long-term evaluation of ecological quality.

Arid areas usually have the natural characteristics of low precipitation, large evapotranspiration, strong wind erosion, and serious water shortage, which makes the carrying capacity of the ecosystem more delicate and vulnerable [31]. In recent years, with the continued disturbance of human activities such as urban expansion, resource development, and climate change [32,33], typical ecological and environmental issues, such as the spatial mismatch between resources and population density and land desertification in arid areas, have been aggravated, with a serious threat to the ecological environment [34]. Therefore, to maintain the stability of economic development in arid regions, serious attention must be given to the protection of ecosystems and to avoid the tendency of comprehensively tilting toward economic development, ignoring the significant role of ecological environment stability in achieving the goals of sustainable development [35,36].

The Xinjiang Uygur Autonomous Region (Xinjiang) is a typical arid inland region which is located in the hinterland of the Eurasian continent, the Northwest border of China [37]. It is a typical arid inland region with extremely low precipitation, as well as large areas of desert and wilderness. In this case, the temperature in Xinjiang is relatively high, while the vegetation coverage and soil moisture are relatively low [38]. At the same time, there is a large gap in the temperature, precipitation, and vegetation coverage between Southern and Northern Xinjiang, which has led to large differences in ecological quality status. Moreover, the seasonal conversion speed in Xinjiang is inconsistent, and there is a large gap in the length of winter in Northern and Southern Xinjiang [39]. Although the accuracy of RSEI has been demonstrated in numerous studies, its performance in large arid areas is still unknown. In this case, this study processed the seasonal ecological quality evaluation in summer and spring to determine the seasonal difference in ecological quality and influencing factors and, ultimately, provide scientific support for ecological management planning in each season. The rest of the paper is arranged as follows: the methods and materials are introduced in Section 2; the results are displayed in Section 3; and the discussion and summaries are presented in Sections 4 and 5, respectively.

## 2. Materials and Methods

### 2.1. Study Area

The Xinjiang Uygur Autonomous Region (Xinjiang) is located in the hinterland of the Eurasian continent and the Northwest border of China (Figure 1). It is the province with the largest land area and the furthest distance from the ocean in China. The Altai Mountains in the North, the Kunlun Mountains in the South, and the Tianshan Mountains lie across the Central part of Xinjiang, forming a unique terrain of "three mountains and two basins". According to this situation, the southern side of the Tianshan Mountains is called Southern Xinjiang, and the north side is called Northern Xinjiang [39]. Xinjiang is located deep inland, surrounded by high mountains. The ocean air flow has difficulty reaching this area, forming an obvious temperate continental climate [40]. Also, there is a large difference in the temperature and precipitation between Northern and Southern Xinjiang; in the summer, the temperature in Southern Xinjiang is obviously higher than in Northern Xinjiang, and the total precipitation in Northern Xinjiang is obviously greater than in Southern Xinjiang [41]. The total length of Xinjiang's land border is more than 5600 km, and it borders eight countries, including Russia, Kazakhstan, Kyrgyzstan, and Tajikistan. Xinjiang was an important passage of the "Ancient Silk Road" in the past, and now it is the place where the second "Eurasian Land Bridge" must pass.

### 2.2. Material Sources

#### 2.2.1. Details of MODIS Data Product

In this study, mainly due to the largeness of the study area and the length of the study period, the MODIS data products were selected as the material sources for the construction of the RSEI, and the total quantity of MODIS data images required for RSEI construction was approximately 7560. Because of the large data requirement of this study, the process of data acquirement was performed based on the Google Earth Engine (GEE) platform's

public data archive (https://developers.google.com/earth-engine/datasets, accessed on 5 May 2023) to select the suitable data products that fit this study's requirement from the MODIS series data product collection from 2000 to 2020 [42].

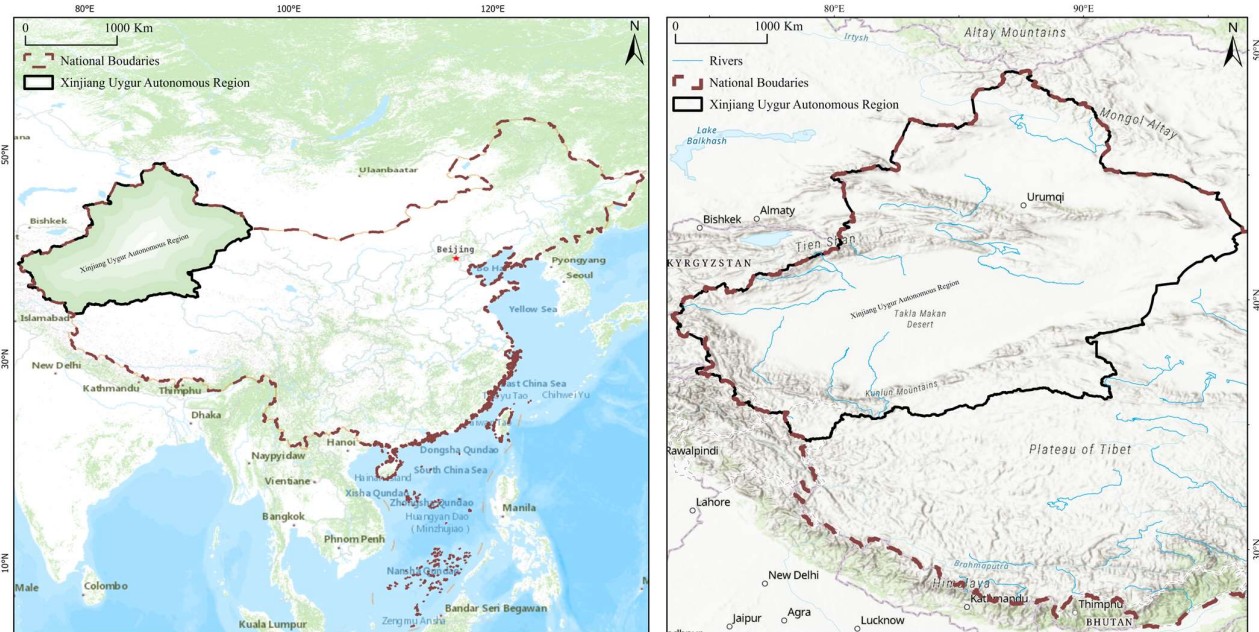

**Figure 1.** Location of this study's area.

For the construction of the RSEI, the MOD13A1 data product was selected to extract the greenness component, and the Normalized Difference Vegetation Index (NDVI) band was used with a spatial resolution of 500 m. The product was integrated by selecting high-quality image pixels through a 16-day period, which could guarantee a real representation of the vegetation coverage extent of this study's area and finely represent the greenness component required by the RSEI. The extraction of the heat component used the MOD11A2 data product, and the LST-Daytime band provided by this product was chosen to represent the LST of the region and was regarded as the heat component [43]. The extraction of the wetness and dryness components involved the MOD09A1 data product. The product afforded an estimated value of the surface spectral reflectance in Terra MODIS from bands 1 to 7, and the pixels were corrected for multiple atmospheric conditions. For each pixel, the product reprocessed 8 days of cyclicality pixel integration (Table 1).

**Table 1.** Details of MODIS data products.

| Products | Time Quantum | | Time Resolution | Platform | Version | Row Number |
|---|---|---|---|---|---|---|
| MOD09A1 | 3.01–5.31 | 6.01–8.31 | 8 days | Terra | V061 | h23v04, v05 |
| MOD11A2 | 3.01–5.31 | 6.01–8.31 | 8 days | Terra | V061 | h24v05, v05 |
| MOD13A1 | 3.01–5.31 | 6.01–8.31 | 16 days | Terra | V061 | h25v05, v05 |

2.2.2. Details of Climatic Data

Considering the special climatic characteristics of this study's area, the most crucial climatic factors of ecological quality in arid land, including temperature and precipitation, were selected for the processing of partial correlation analysis with the aim of determining the relationship between climatic factors and ecological quality. For the data source of the partial correlation analysis, the monthly data sets of temperature and precipitation of this study's area from 2000 to 2020, both in the summer and spring, were acquired from

the National Qinghai–Tibet Plateau Data Center (http://data.tpdc.ac.cn/, accessed on 6 May 2023) (Table 2).

**Table 2.** Basic information on climatic data.

| Data Sets | Time Quantum | | Spatial Resolution | Data Source |
|---|---|---|---|---|
| Precipitation | 3.01–5.31 | 6.01–8.31 | 1000 m | (http://data.tpdc.ac.cn/) |
| Temperature | 3.01–5.31 | 6.01–8.31 | 1000 m | (http://data.tpdc.ac.cn/) |

*2.3. Methodology*

In this study, the ecological environmental quality in Xinjiang from 2000 to 2020 was assessed by analyzing the spatiotemporal distribution and variation tendency of the RSEI both in the summer and spring with a spatial resolution of 1000 m. Meanwhile, the significance and persistency evaluation system of ecological environmental quality was established to systematically analyze the changing trend of ecological environmental quality. Finally, the partial correlation analysis was processed between ecological quality and climatical factors, including temperature and precipitation, to determine the correlation pattern of climatic factors with ecological environment quality (Figure 2).

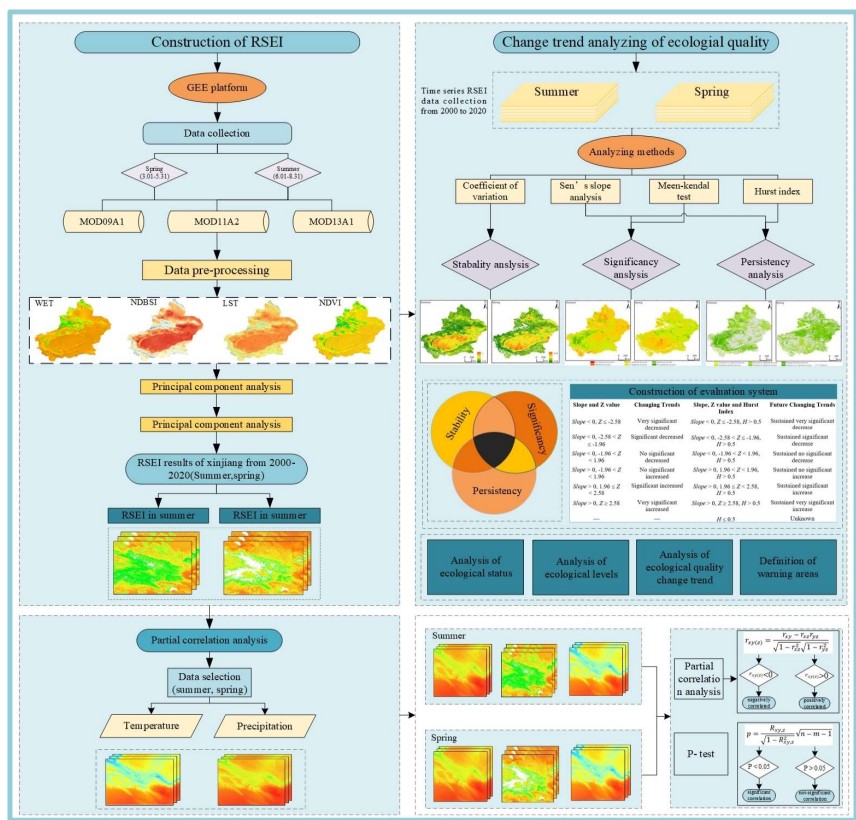

**Figure 2.** The general framework of this study.

2.3.1. Construction of RSEI

According to the general construction principles of the RSEI and the scientific findings of previous studies related to ecological environment assessment, within the multiple natural related indices that reflect the eco-environmental quality, greenness, wetness, heat, and dryness are regarded as the four crucial components which directly correlated to the living quality of the human habitat. Based on this, the above four components are frequently used to evaluate the quality of ecosystems by scholars individually or jointly,

and the above four were successfully integrated into the RSEI for a more comprehensive and scientific evaluation of ecological environmental quality.

In this study, in order to ensure that the RSEI was able to reflect the real condition of ecological environmental quality in this study's area in a more effective and accurate way, it was necessary to take effect from large areas of water bodies in all forms, including lakes, rivers, and glaciers, into consideration. With the aim of avoiding the negative effects of water in waterbodies to the ultimate result of the RSEI, the Modified Normalized Difference Water Body Index (MNDWI) was extracted from original MODIS data products as the representation of water bodies in this study's area, and the MNDWI used the water mask and refreshed it each year to make sure the wetness component truly represented the moisture of land surface [44].

Greenness: The NDVI is undoubtedly the most widely applied vegetation index in various studies of the need for visualization and evaluation of the vegetation coverage in specified areas, mainly because the index effectively reflects crucial indices such as the Vegetation Biomass, Leaf Area Index (LAI), and vegetation coverage. For this reason, the NDVI index represented the greenness component and was used for the construction of the RSEI.

Wetness: The wetness component was characterized by the MOD09A1 product by processing the tasseled cap transformations on each band of the product, which were nearly related and able to reflect the vegetation and soil moisture condition of the region.

Heat: The LST is the index that could most directly represent the heat component; in this case, the LST-Daytime band, provided by the MOD11A2 data product, was chosen to represent the heat component for the construction of the RSEI.

Dryness: The Index-Based Built-up Index (IBI) is a widely accepted index that is usually chosen to reflect the drought extent of the region; meanwhile, a major part of bare ground in the regional environment still exists, which could also generate a "drying" effect on the land surface. Therefore, the dryness index could be more comprehensive when it is synthesized from the integration of IBI and the soil index (SI). For this reason, the MOD09A1 data product was selected for the extraction of (NDBSI), which was established by Hu and Xu to represent the dryness component.

The above four components were extracted, and the normalization processes on each component were completed to control the dimension range. To successfully construct the RSEI and integrate the required components, which were extracted individually, statistical method principal component analysis (PCA) was performed. According to the statistical principles of PCA, within the results of four principal components, the eigenvalues of the PC1 synthesize the most important information of each component together. For this reason, the results of PC1 were chosen to describe the primitive remote sensing ecological index ($RSEI_0$). To facilitate the process of measurement and comparison, the $RSEI_0$ was further normalized based on the following equation [30]:

$$RSEI = (RSEI_0 - RSEI_{0\_min})/(RSEI_{0\_max} - RSEI_{0\_min}) \tag{1}$$

where RSEI represents the ultimate results of remote sensing ecological index. Based on the general principles of RSEI, the value range is between [0, 1]. If the result is close to 1, then the ecological quality of the region is better, and if the value is close to 0, then the situation is exactly opposite.

### 2.3.2. From–to Change Analysis

The from–to change analysis is an image calculation methodology used to analyze the variation trend of raster data between two time points. It is helpful for visualizing the change trend in raster data with the help of mathematical calculation. In this study, the reclassified RSEI level data in 2000, 2010, and 2020 were chosen for processing for the from–to analysis from 2000 to 2010, from 2010 to 2020, and from 2000 to 2020 [45–47].

### 2.3.3. Time-Series Tendency Analyzing Methods

The time-series variation tendency analyzing methods applied to this study included the rescaled range method (R/S), the variant coefficient method CV (coefficient of variation), Sen's slope trend analysis, and the Mann–Kendall trend test (M–K test) for processing the analyzation for the RSEI of this study's area from 2000 to 2020 [48].

1.    Sen's Slope Trend Analysis

The Sen's slope trend analysis (Sen's) is a classic statistical method that is usually applied to evaluate the overall variation trend in original data over a specified time period. According to the principles of the method, the maximal advantage of this method is that it can effectively avoid the effect of missing and outlier values on the analysis result while processing. Based on the fundamental operating principles of Sen's, the whole process could be divided into the following two steps. First, the calculation of the slope between all adjacent time-point data in the time series is needed. Second, the median value of the slope is chosen as the numerical result, which is able to indicate the changing trend of the original raster data. The basic mathematical function of Sen's slope analysis is as follows [24,49]:

$$slope = median\left(\frac{RSEI_j - RSEI_i}{j - i}\right), \forall j > i \tag{2}$$

where *median* represents the median function, and the $RSEI_j$ and $RSEI_i$ are the values, which were, respectively, observed at the time point $j$ and $i$ in the whole time series. When the results of Sen's slope < 0, it means that the ecological quality tends to decrease, and when Sen's slope > 0, it means a tendency to increase.

2.    Mann–Kendall Trend Test

The Mann–Kendall trend test (M–K test) is a nonparametric trend testing method that is regularly applied in combination with Sen's to reveal the significancy of the variation tendency in the original time-series data. As with the similar characteristics of Sen's, the M–K test is also instinctively insensitive to the missing and outlier values existing in the original data. Its general distribution statistic, $S$, was calculated via the following equation [24,25]:

$$S = \sum_{i=1}^{n-1} \sum_{j=i+1}^{n} sgn\left(RSEI_j - RSEI_i\right) \tag{3}$$

where $RSEI_j$ and $RSEI_i$ are the RSEI values observed at the time point $j$ and $i$, respectively, and the *sgn* is a conforming function, the expression of which is presented below:

$$sgn(x_j - x_i) = \begin{cases} 1, x_j - x_i > 0 \\ 0, x_j - x_i = 0 \\ -1, x_j - x_i < 0 \end{cases} \tag{4}$$

According to Equation (4), it can be seen that the value of the *sgn* function varied from $-1$ to 1 based on the value of $(RSEI_j - RSEI_i)$. The variance, $Var(S)$, of $S$ can be calculated through the equation presented below:

$$Var(S) = \frac{n(n-1)(2n+5)}{21} \tag{5}$$

According to the results of $S$ and $Var(S)$, which were calculated via Equations (3) and (5), the standard orthogonal statistical distribution, $Z$, of the M–K test can be calculated used Equation (6):

$$Z = \begin{cases} \frac{S-1}{\sqrt{Var(S)}}, & S > 0 \\ 0, & S = 0 \\ \frac{S+1}{\sqrt{Var(S)}}, & S < 0 \end{cases} \tag{6}$$

Referring to previous related studies, this study used 95% and 99% confidence intervals, which indicates that the changing trend of original data series is very significant when $Z \geq 2.58$ and $Z \leq -2.58$, and it is significant when $1.96 \leq Z < 2.58$ and $-2.58 < Z \leq -1.96$; otherwise, the changing trend is regarded not significant. In this study, to vividly present the changing trend of RSEI, the results of Sen's slope analysis and the M–K test were superimposed and classified the changing trends of the RSEI into six types for the change trend significancy analysis (Table 3).

**Table 3.** Summary of the calculation methods for each ecological factor.

| Components | Index | Functions | References |
|---|---|---|---|
| Greenness | NDVI | $NDVI = \frac{NIR - RED}{NIR + RED}$ | [27,30] |
| Wetness | WET | $Wet = H1\rho_1 + H2\rho_2 + H3\rho_3 + H3\rho_4 - H5\rho_5 - H6\rho_6 - H7\rho_7$ | [45] |
| Heat | LST | $LST = 0.02LST_0 - 273.15$ | [46] |
| Dryness | NDBSI | $IBI = \frac{\left\{\frac{2\rho_6}{\rho_6 + \rho_2} - \left[\frac{\rho_2}{\rho_2 + \rho_1} + \frac{\rho_4}{\rho_4 + \rho_6}\right]\right\}}{\left\{\frac{2\rho_6}{\rho_6 + \rho_2} + \left[\frac{\rho_2}{\rho_2 + \rho_1} + \frac{\rho_4}{\rho_4 + \rho_6}\right]\right\}}$ $SI = \frac{[(\rho_6 + \rho_1) - (\rho_2 + \rho_3)]}{[(\rho_6 + \rho_1) + (\rho_2 + \rho_3)]}$ $NDBSI = (IBI + SI)/2$ | [29] |

Note: where $\rho_1 \ldots \rho_7$ represent the bands of the MOD09A1 data products from sur_refl_b01 to sur_refl_b07, and $H1$~$H7$ represent the moisture coefficients of each band, and the value of coefficients in order are $H1 = 0.1147$, $H2 = 0.2489$, $H3 = 0.2408$, $H4 = 0.3132$, $H5 = 0.3122$, $H6 = 0.6416$, and $H7 = 0.5087$.

3. Hurst Index

The Hurst index is an analyzing method based on the rescaled polar deviation (R/S), which can effectively reflect the autocorrelation of data in a time series, especially within the hidden long-term change trend. Based on this privilege, the Hurst index is able to efficiently represent the persistency of original data variation in a time series. It reflects the correlation of changing trends between two time points, which means the previous variation trends could affect the present variation trends, and the present variation trends could affect the future variation trends [50]. In this case, the Hurst index was chosen to calculate the statistics of the future variation tendency of the RSEI. The general mechanism of the Hurst index is described below [51,52]:

Set a time series as: $\{RSEI_{(t)}, t = 1, 2, \ldots n\}$, and for any integer $\alpha \geq 1$, the average series is shown in the equation presented below:

$$\overline{RSEI}_{(t)} = \frac{1}{T}\sum_{t=1}^{T} RSEI_{(t)} \quad t = 1, 2, \cdots, n \tag{7}$$

The cumulative deviation was calculated by Equation (8):

$$X(t, \alpha) = \sum_{t=1}^{t} \left(RSEI(t) - \overline{RSEI}_\alpha\right) \quad 1 \leq t \leq \alpha \tag{8}$$

The value scope of excessive diversity, $R(\alpha)$, was defined as in Equation (9):

$$R(\alpha) = \max_{1 \leq t \geq \tau} X(t, \alpha) - \min_{1 \leq t \geq \tau} X(t, \alpha) \quad \alpha = 1, 2, \cdots, n \tag{9}$$

Calculation of the standard deviation, $S(\alpha)$, is presented in Equation (10):

$$S(\alpha) = \left[\frac{1}{\alpha}\sum_{t=1}^{\alpha} \left(LST(t) - \overline{LST}_\alpha\right)^2\right]^{1/2} \quad \alpha = 1, 2, \cdots, n \tag{10}$$

Finally, the Hurst index was calculated by Equations (11) and (12):

$$\frac{R(\alpha)}{S(\alpha)} = (c\alpha)^H \tag{11}$$

$$log(R/S)_n = a + H \times log(n) \tag{12}$$

where $H$ represents the result of the Hurst index, $c$ represents the relation constants, and $a$ represents the interception. The $H$ values can be calculated by Equation (12) using the least-squares methods. According to the principles of the Hurst index, when $H > 0.5$, the future variation tendency of the RSEI is similar to the current condition; when $H < 0.5$, the future variation tendency of the RSEI may be opposite to the current condition; and when $H = 0.5$, the future changing trend of the RSEI is uncertain. Based on these principles, in this study, $H = 0.5$ was defined as the value in which the future change tendency is unknown. Also, for a better understanding of the future changing trend of the RSEI, the evaluation system of the changing trend of RSEI was constructed by synthesizing the results of the Hurst index, Sen's slope analysis, and the M–K test, as shown in Table 4.

**Table 4.** RSEI change trend significancy and persistency evaluation system.

| Slope and M–K | Significancy of RSEI Trend | Slope, M–K and Hurst | Persistency of RSEI |
|:---:|:---:|:---:|:---:|
| $S < 0, Z \leq -2.58$ | Very significant decreased | $S < 0, Z \leq -2.58, H > 0.5$ | Sustained very significant decrease |
| $S < 0, -2.58 < Z \leq -1.96$ | Significant decreased | $S < 0, -2.58 < Z \leq -1.96, H > 0.5$ | Sustained significant decrease |
| $S < 0, -1.96 < Z < 1.96$ | Slightly decreased | $S < 0, -1.96 < Z < 1.96, H > 0.5$ | Sustained slightly significant decrease |
| $S > 0, -1.96 < Z < 1.96$ | Slightly increased | $S > 0, 1.96 < Z < 1.96, H > 0.5$ | Sustained slightly increase |
| $S > 0, 1.96 \leq Z < 2.58$ | Significant increased | $S > 0, 1.96 \leq Z < 2.58, H > 0.5$ | Sustained significant increase |
| $S > 0, Z \geq 2.58$ | Very significant increased | $S > 0, Z \geq 2.58, H > 0.5$ | Sustained very significant increase |
| — | — | $H \leq 0.5$ | Unknown |

#### 2.3.4. Partial Correlation Analysis

Partial correlation analysis was used to determine the relationship between the specified influencing factors and the dependent variable. According to this characteristic, the method is widely used to separate the relationship between two closely related variables from the confounding effects with multiple correlated variables. In this case, we aimed to find the relationship between ecological quality and the essential climate element in each pixel. The partial correlation analysis was processed to the data, including the RSEI, precipitation, and temperature. According to the principles of this method, the values of each partial correlation coefficient are located in [0, 1]. When the values of the partial correlation coefficient are equal or infinitely close to 1, this indicates that the two variables are perfectly correlated. Conversely, when the absolute value of the partial correlation coefficient is equal or infinitely close to 0, this indicates that the two variables are completely uncorrelated. Additionally, if the value of the partial correlation coefficient is positive, then the two variables are positively related; if the value of the partial correlation coefficient is negative, then the two variables are negatively related. The function of partial correlation analysis is as follows [53,54]:

$$r_{xy(z)} = \frac{r_{xy} - r_{xz}r_{yz}}{\sqrt{1 - r_{xz}^2}\sqrt{1 - r_{yz}^2}} \tag{13}$$

where $r_{xy(z)}$ indicates the partial correlation coefficient between variable $x$ and variable $y$ after fixing variable $z$; and $r_{xy}$, $r_{xz}$, and $r_{yz}$ represent the correlation coefficients between variable $x$ and variable $y$, between variable $x$ and variable $z$, and between variable $y$ and

variable $z$, respectively. Significance tests for bias correlation coefficients were performed using the $p$-test, and the statistic was calculated as follows:

$$p = \frac{R_{xy,z}}{\sqrt{1 - R_{xy,z}^2}} \sqrt{n - m - 1} \tag{14}$$

where $R_{xy,z}$ represents the partial correlation coefficient; $n$ is the number of samples, and $m$ is the degree of freedom. If $p > 0.05$, then the correlation is not obvious; if $p < 0.05$, then the correlation is obvious.

## 3. Results and Analysis

### 3.1. Ecological Quality Status in Xinjiang

According to the RSEI results of Xinjiang in the summer and spring from 2000 to 2020, the comparable analysis of ecological quality was processed as shown in Figures 3 and 4. Whether in the summer or spring, the low RSEI values were concentrated in the southwestern, eastern, and southeastern parts of this study's area, which were mostly dominated by desert and wilderness. High RSEI values were concentrated in the areas around the oasis, and the RSEI gradually improved from the inside to the outside. During the summer, in Northern Xinjiang, the high RSEI values were mainly distributed in the Yili River Basin and Irtysh River Basin. In Southern Xinjiang, the high RSEI values were mainly distributed in arid oasis areas such as the Hotan Oasis, Kashgar Oasis, and Aksu Oasis around the Tarim Basin. During the spring, the main distribution characteristics of the RSEI in Xinjiang were almost the same as in the summer, with slight seasonal differences. Unlike the summer, in Northern Xinjiang, the high RSEI values were mainly distributed in the Yili River Basin. In Southern Xinjiang, the high RSEI values were mainly distributed in the Kashgar Oasis, Aksu Oasis, and Hotan Oasis, and the mean value of the RSEI continued to increase.

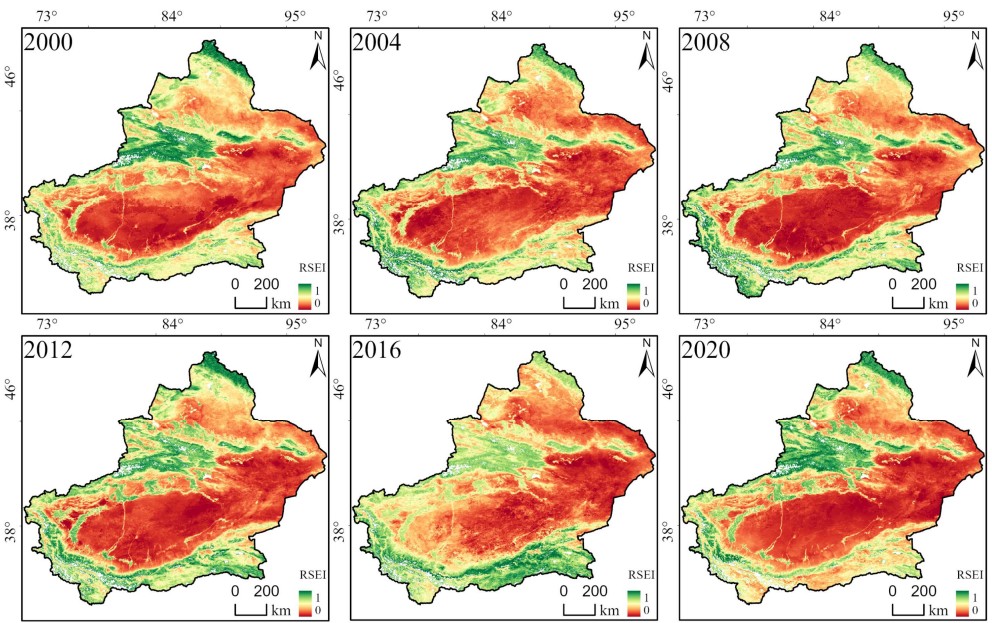

**Figure 3.** RSEI changes in Xinjiang in summer.

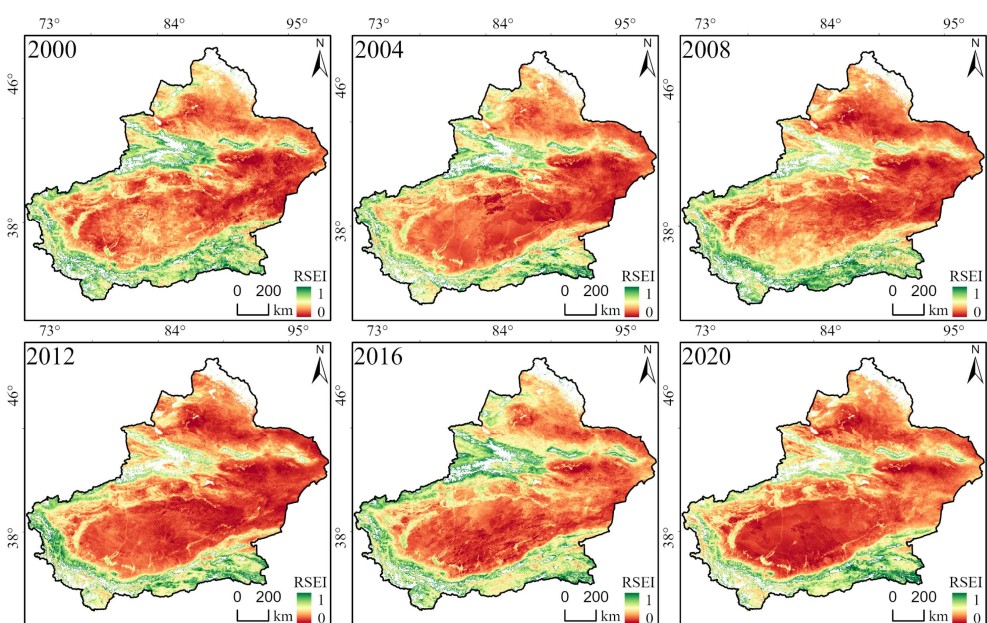

**Figure 4.** RSEI changes in Xinjiang in spring.

For a more detailed analysis of the RSEI changes, the mean values of the RSEI from 2000 to 2020, both in the summer and spring, were extracted (Figures 5 and 6). The contribution of the mean value shows that the contribution characteristics in the summer and spring were approximately similar; the high mean values were mostly distributed in the areas around the Tarim Basin, Yili River Basin, and Irtysh River Basin in the summer. The high mean values were distributed around the Tarim Basin and Yili River Basin in the spring. Meanwhile, the mean value in the summer was obviously higher than in the spring.

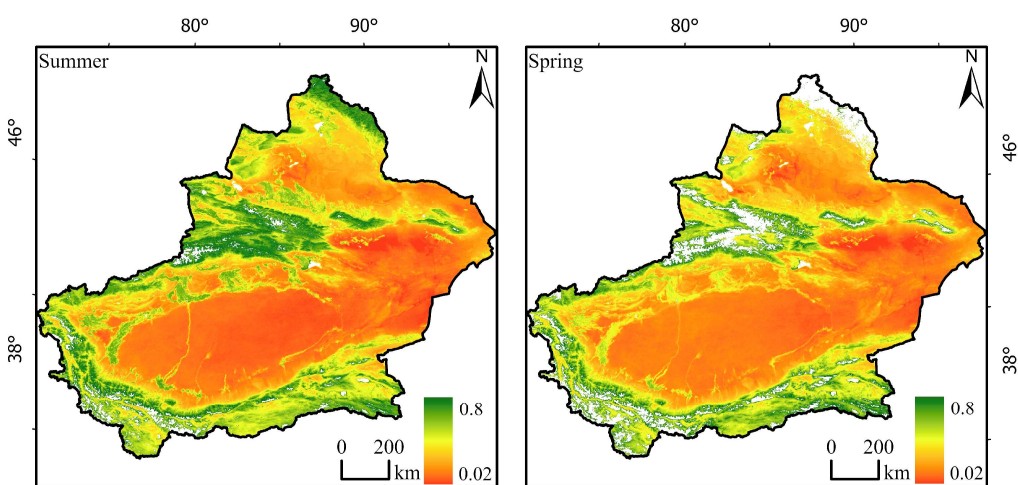

**Figure 5.** Spatiotemporal changes in RSEI mean value in Xinjiang.

### 3.2. Ecological Quality Variations in Xinjiang

Based on the overall consideration of the actual situation of this study's area and the ecological environment quality grading standards issued by the Ministry of Ecology and Environment, this study decided to divide the RSEI value of Xinjiang both in the summer and spring into five levels with an interval of 0.2 using the natural breakpoint method. According to this classification criteria, the RSEI values of 0–0.2, 0.2–0.4, 0.4–0.6, 0.6–0.8, and 0.8–1 were defined as poor, fair, moderate, good, and excellent, respectively.

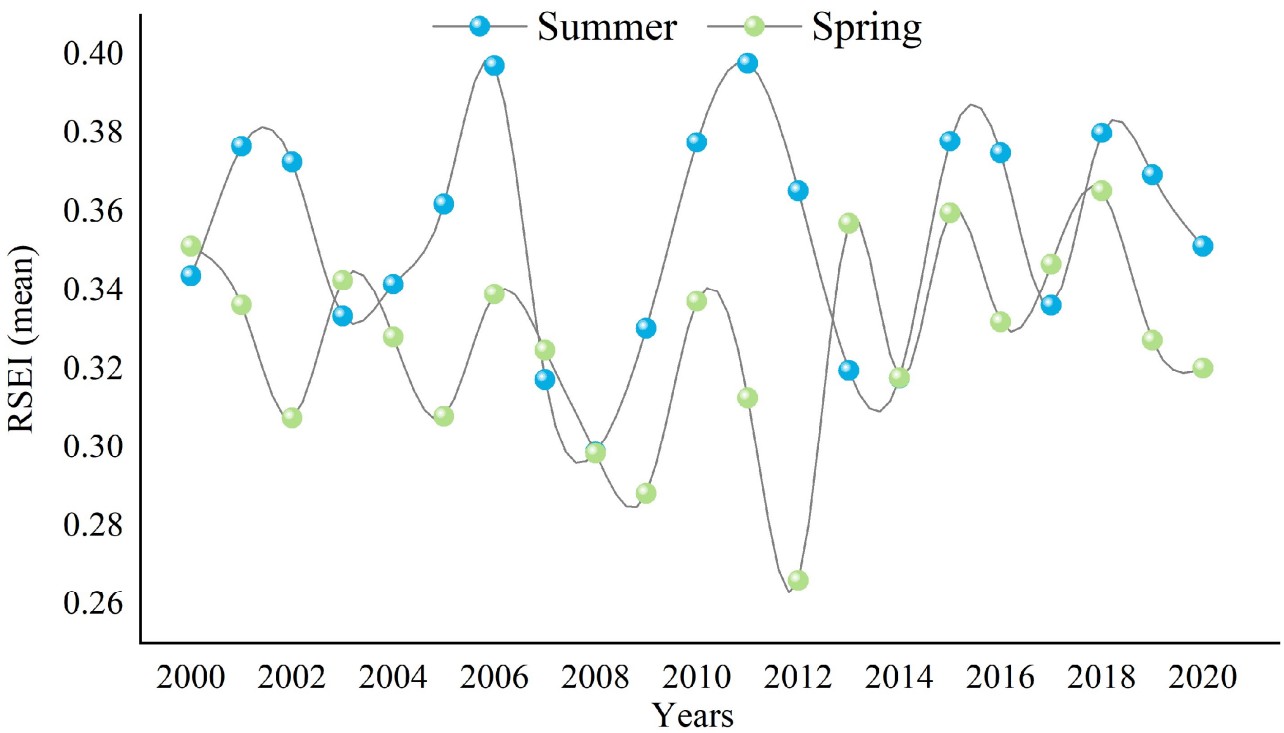

**Figure 6.** RSEI mean value changes in Xinjiang in summer and spring.

The spatiotemporal distribution of RSEI levels both in summer (Table 5 and Figure 7) and spring (Table 6 and Figure 8) from 2000 to 2020 was as follows: in the summer, the level of RSEI was increased from the inside to the outside. Poor RSEI was mainly distributed in the southwestern and eastern regions of this study's area; in addition, the poor area stopped decreasing and reached the minimum in 2015. The fair and moderate RSEIs were mainly distributed in the oasis around the Tarim Basin in Southern Xinjiang and around the Junggar Basin in Northern Xinjiang. The good and excellent areas were mostly in Northern Xinjiang and were slightly increased in Southern Xinjiang. The whole area of good and excellent increased from 175,726 km² and 7296 km² in 2000 to 224,192 km² and 12,023 km² in 2020, respectively. In the spring, the fluctuation of the poor area was significant, and the area increased from 261,162 km² in 2000 to 437,330 km² in 2020. The whole fair area increased from 747,732 km² in 2000 to 641,650 km² in 2020. The moderate and fair areas also gradually decreased, while the excellent area gradually increased from 7296 km² in 2000 to 12,391 km² in 2020.

**Table 5.** RSEI level distribution of Xinjiang in summer (Unit: area km², percentage %).

| RSEI Level | 2000 | | 2005 | | 2010 | | 2015 | | 2020 | |
|---|---|---|---|---|---|---|---|---|---|---|
| | Area | Per | Area | Per | Area | Per | Area | Per | Area | Per |
| Poor | 451,699 | 27.21 | 416,646 | 25.09 | 360,244 | 21.7 | 283,948 | 17.1 | 467,443 | 28.16 |
| Fair | 564,174 | 33.98 | 570,044 | 34.34 | 587,047 | 35.36 | 647,259 | 38.99 | 558,994 | 33.67 |
| Moderate | 395,478 | 23.82 | 376,330 | 22.67 | 360,743 | 21.73 | 439,402 | 26.47 | 331,656 | 19.98 |
| Good | 175,726 | 10.58 | 225,245 | 13.56 | 267,007 | 16.08 | 216,934 | 13.06 | 224,192 | 13.5 |
| Excellent | 7296 | 0.43 | 6097 | 0.37 | 19,314 | 1.16 | 6777 | 0.41 | 12,023 | 0.72 |

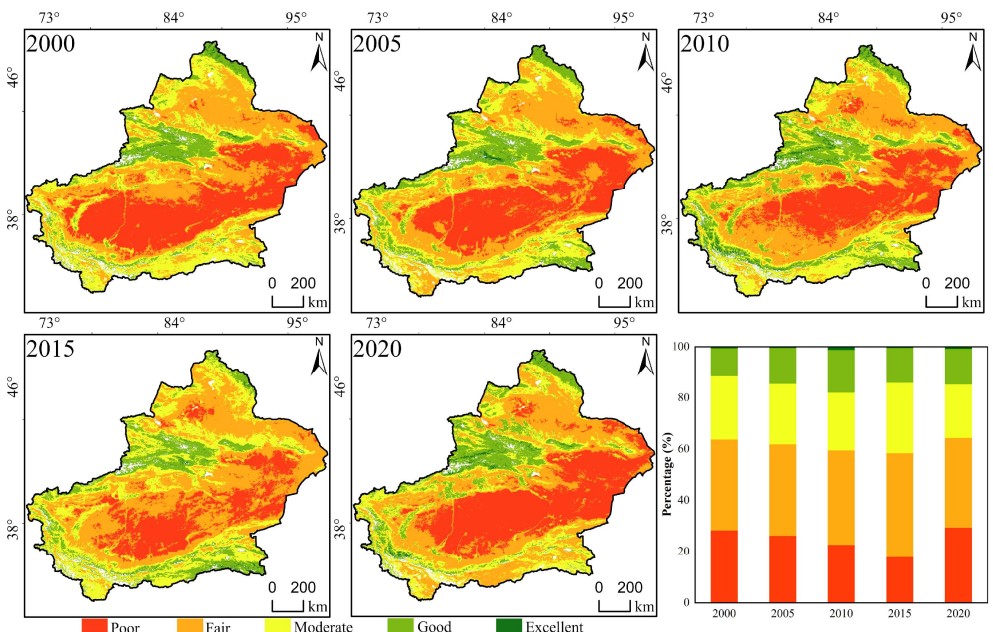

**Figure 7.** Ecological quality distribution of Xinjiang in summer.

**Table 6.** RSEI level distribution of Xinjiang in spring (Unit: area km², percentage %).

| RSEI Level | 2000 | | 2005 | | 2010 | | 2015 | | 2020 | |
|---|---|---|---|---|---|---|---|---|---|---|
| | Area | Per | Area | Per | Area | Per | Area | Per | Area | Per |
| Poor | 261,162 | 15.73 | 450,964 | 27.16 | 263,810 | 15.89 | 285,237 | 17.18 | 437,330 | 26.34 |
| Fair | 747,732 | 45.04 | 659,059 | 39.7 | 781,807 | 47.09 | 681,472 | 41.05 | 641,650 | 38.65 |
| Moderate | 317,446 | 19.12 | 295,009 | 17.77 | 343,968 | 20.72 | 330,604 | 19.91 | 285,769 | 17.21 |
| Good | 155,907 | 9.39 | 80,386 | 4.84 | 99,369 | 5.94 | 175,696 | 10.58 | 112,571 | 6.78 |
| Excellent | 7978 | 0.48 | 4574 | 0.27 | 1130 | 0.06 | 16,779 | 1.01 | 12,931 | 0.77 |

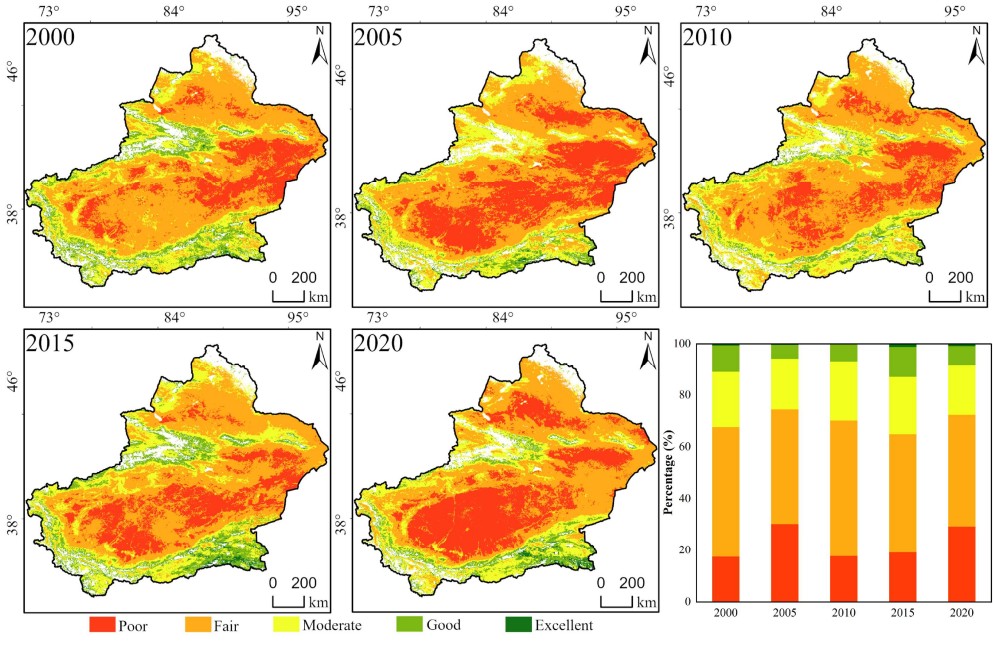

**Figure 8.** Ecological quality distribution of Xinjiang in spring.



### 3.3. From–to Analysis of RSEI Level

In order to analyze the transformation trend of the RSEI level in this study's area, the from–to analysis from 2000 to 2010, from 2010 to 2020, and the overall change trend from 2000 to 2020 were processed, respectively. The results of the from–to analysis are shown in Figure 9. In the summer, from 2000 to 2010, the RSEI level change trend was mainly dominated by a large area with an unchanged trend and the transformation from poor and fair to moderate and good, which means that in these areas, the ecological level of RSEI tended to rise; this change trend was obvious around the Taklamakan Desert, Junger Basin, and the southeastern part of this study's area. From 2010 to 2020, there was an obvious increase in the transformation from good to moderate and good to excellent along the Kunlun Mountains. The overall changing trend from 2000 to 2020 indicates that the RSEI level of this study's area was relatively stable in the last 21 years, with a large unchanged area and a large area that transformed from poor and fair to moderate and good; this change trend was more obvious in the oasis around the Tarim Basin and the area around the Kumtag Desert. Also, the moderate area that transformed into good and excellent was distributed in most parts of this study's area. In the spring, from 2000 to 2010, the change trend of the RSEI level in this study's area was also dominated by a large unchanged area, with the transformation from poor and fair to higher levels, such as moderate and good. From 2010 to 2020, a large unchanged area transformed from poor and fair to higher levels, and the moderate area that changed to poor and fair in the edge of the Taklamakan Desert was replaced by the transformation of poor and fair to moderate and good. The overall change trend results of the RSEI level from 2000 to 2020 in the spring show that the large unchanged area was distributed in this study's area. The positive transformation from poor and fair to higher levels also occupied a large area that extended from the southwestern part to the southeastern part of this study's area.

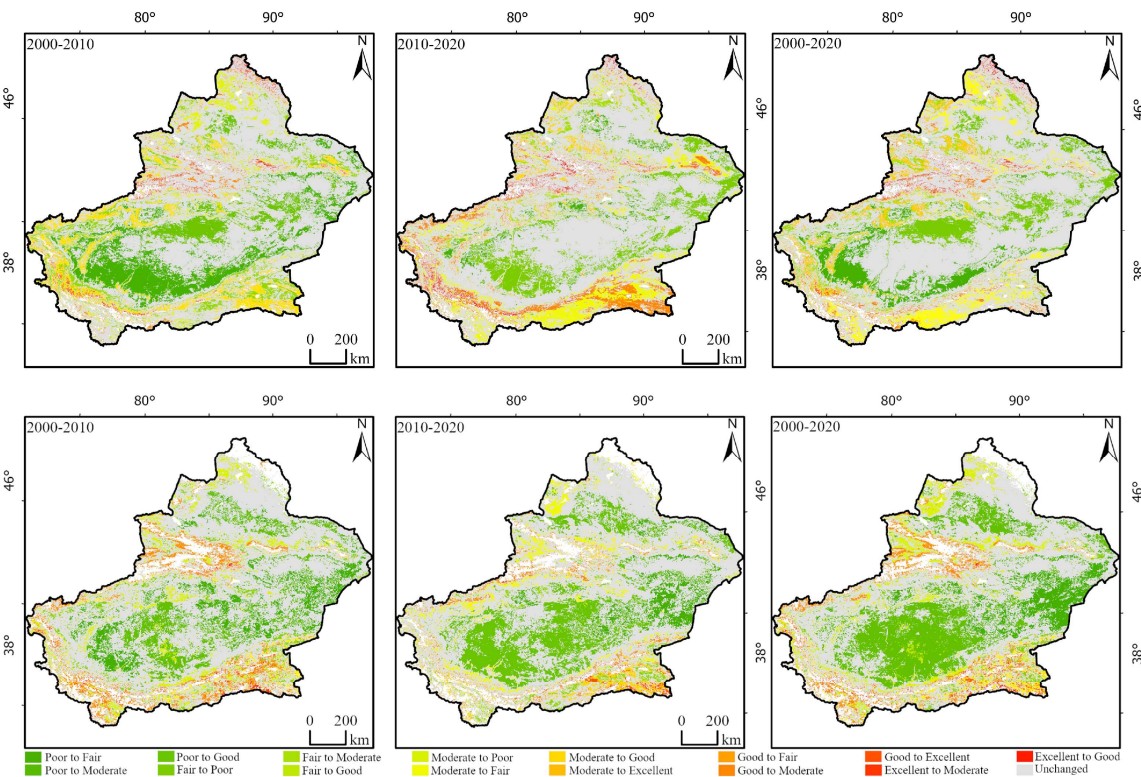

**Figure 9.** RSEI level changes in this study's area.

*3.4. Time-Series Evolution Analysis of the Ecological Environment*

In this study, based on the original RSEI results of the study area, the spatial measurement and time-series evolution analysis were processed on the ecological environment quality of Xinjiang. The result of the CV from 2000 to 2020 shows (Figure 10) that both in the summer and spring, the CV values were concentrated between 0 and 0.02, and the percentages were, respectively, 64.17% in the summer and 56.2% in the spring. This result indicates that the fluctuation of ecological quality in this study's area within the study period was small and stable. In the summer, the high CV values were concentrated in the Taklamakan Desert, which is located in the southwestern part, and the Kumtag Desert in the southeastern part of this study's area. In the spring, the high CV values were concentrated in the Taklamakan Desert, Kumtag Desert, and Ta'e Basin. The regions mentioned above are mostly desert or wilderness, which are sensitive to seasonal changes and affected by human activities, leading to large RSEI fluctuations.

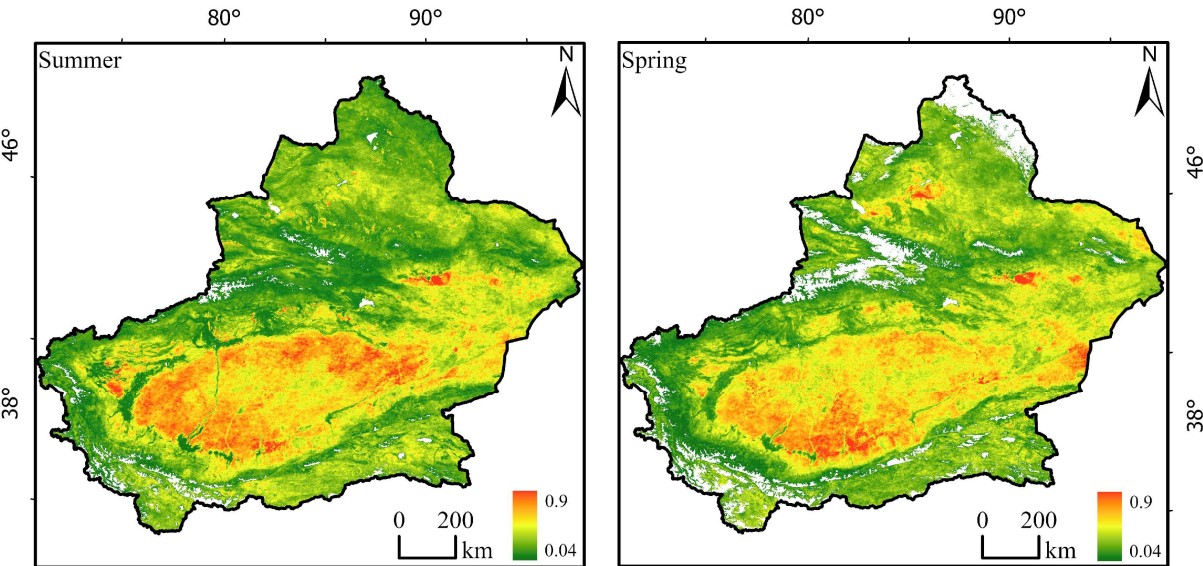

**Figure 10.** Coefficient of variation in this study's area.

According to the integration results of Sen's slope and the M–K test, the distribution of ecological quality change trend significancy in this study's area is shown in Figure 11. In the summer, from 2000 to 2020, the change trends of this study's area were dominated by slightly decreased and slightly increased values. The slightly decreased area was mostly concentrated in Northern and Eastern Xinjiang. Meanwhile, the area with a very significant decrease was concentrated in the eastern and northeast parts of this study's area. The large area with no significant increase was concentrated in the oasis around the Tarim Basin. This indicates that, in the last 21 years, the ecological quality of Southern Xinjiang has significantly improved while the Northern Xinjiang area has remained stable with a small area that has significantly decreased. In the spring, the areas of slightly increased and very significantly increased change trends were mostly dominated in Northern Xinjiang and the part of Southern Xinjiang, which is close to the Tianshan mountains. Meanwhile, there were small areas with very significant decreases in Southern Xinjiang, with a large area of no slightly decreased change trends.

According to the integration results of Sen's slope, the Mann–Kendall test, and the Hurst index, the distribution of ecological quality change trend and persistency in this study's area are shown in Figure 12. In the summer, from 2000 to 2020, the change trend of ecological quality was mainly dominated by an unknown factor, which was distributed in most parts of this study's area. Also, there were small areas of sustained, very significant decreases in the Northern Xinjiang, mainly around the Ta'e Basin. The areas of sustained

increase, including slight and significant increases, were distributed in the whole study area. The sustained increasing trend in Southern Xinjiang was more obvious in the oasis around the Tarim Basin. In the spring, the area affected by an unknown factor still dominated the whole study area and was mostly concentrated in Southern Xinjiang. There were small areas of sustained, very significant decreases in the middle of the Taklamakan Desert, while there was a large area of sustained increase, including very significant and significant increases, in the middle and eastern parts of this study's area.

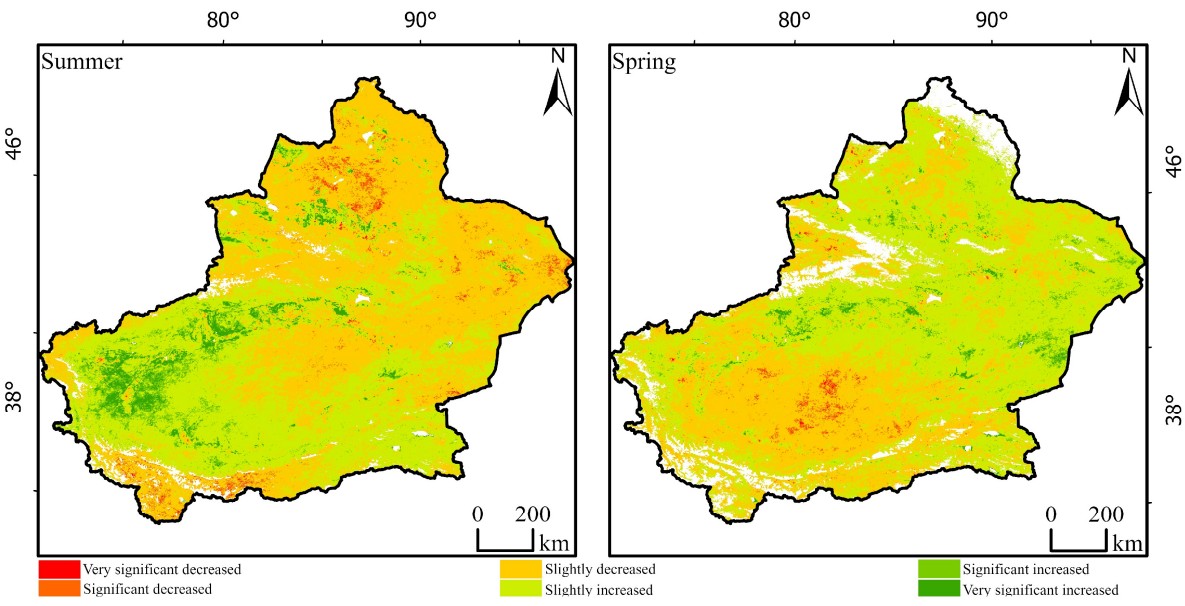

**Figure 11.** Temporal evolution trend analysis in this study's area.

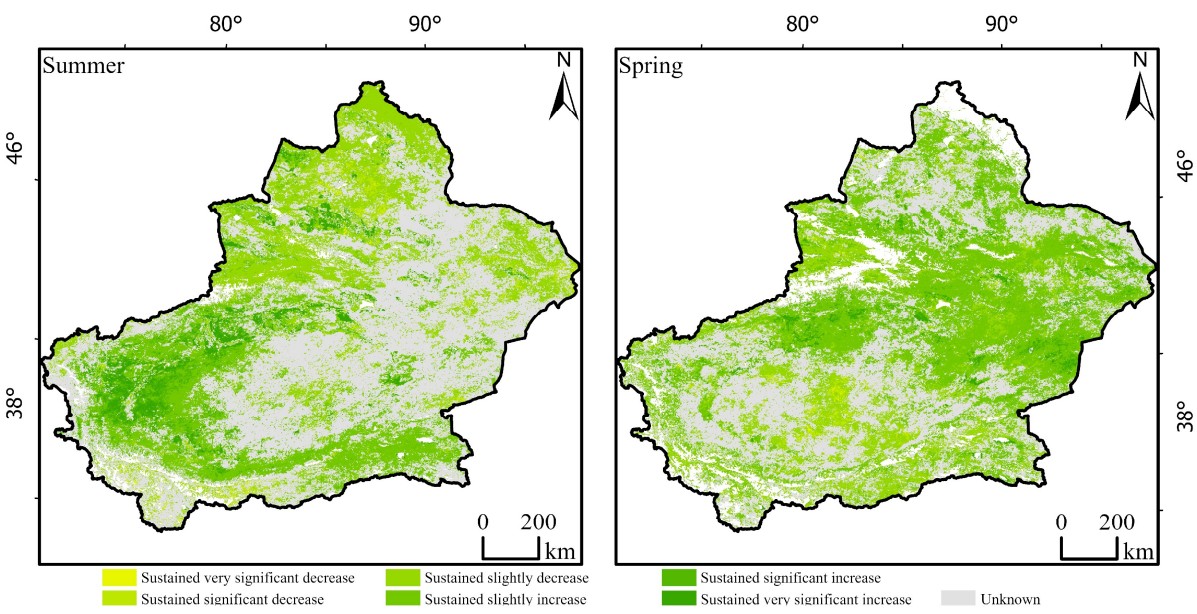

**Figure 12.** Temporal trend persistency analysis of RSEI.

### 3.5. Partial Correlation Analysis

The spatial distribution of the partial correlation coefficient between the RSEI and climatic factors selected in this study, both in the summer and spring, are shown in Figures 13 and 14. In the summer, the range of the partial correlation of the RSEI between precipitation and temperature was from −0.74 to 0.87 and from −0.73 to 0.90, respec-

tively, with mean values of 0.13 and 0.038. The partial correlation result in the summer between the RSEI and precipitation indicates that in the area around the Tarim Basin, the precipitation was negatively correlated with the RSEI; however, this was not obvious. The small areas in the Kashgar Oasis, Hotan Oasis, and Hami Oasis presented obvious positive correlations. The results of the temperature and RSEI presented in the Yili River Basin, the southwestern part of the Tarim Basin, and the southeastern part of this study's area, including the Hami and Turpan Oasis, were negatively not correlated. In the northwest part of the Tarim Basin, the temperature was positively correlated with the RSEI, and this was obvious according to the significancy test. In the spring, the partial correlation result of the RSEI and precipitation was dominated by a positive correlation, mostly in the southwestern part of this study's area, including the Hotan Oasis and Kashgar Oasis, and this correlation was positive. At the same time, there were scattered small areas of negative correlations around the Juggar Basin, Kumtag Desert, and Kurbantunggut Desert, but the correlations were insignificant. The correlation between the RSEI and temperature presented large areas of positive correlation in the northern part of the Tarim Basin, Yili River Basin, and Junggar Basin, and the correlation trend was significant. Also, the negative correlations, mostly in the southwestern part of the Tarim Basin, Kumtag Desert, Turpan Basin, and Kurbantunggut Desert, were insignificant.

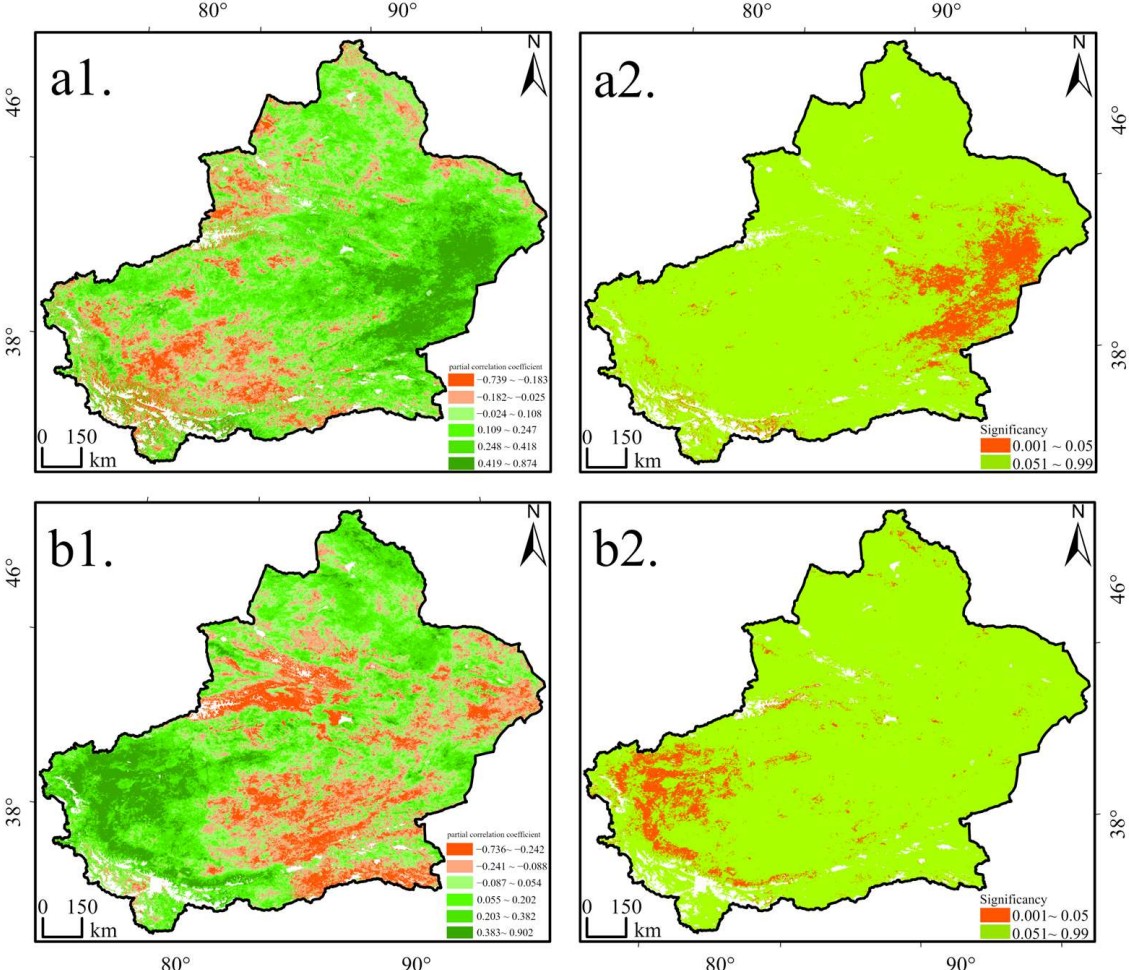

**Figure 13.** Partial correlation analysis results in summer. Note: (**a1**) is the partial correlation results of precipitation, (**a2**) is the significancy results of precipitation, (**b1**) is the partial correlation results of temperature, (**b2**) is the significancy results of temperature.

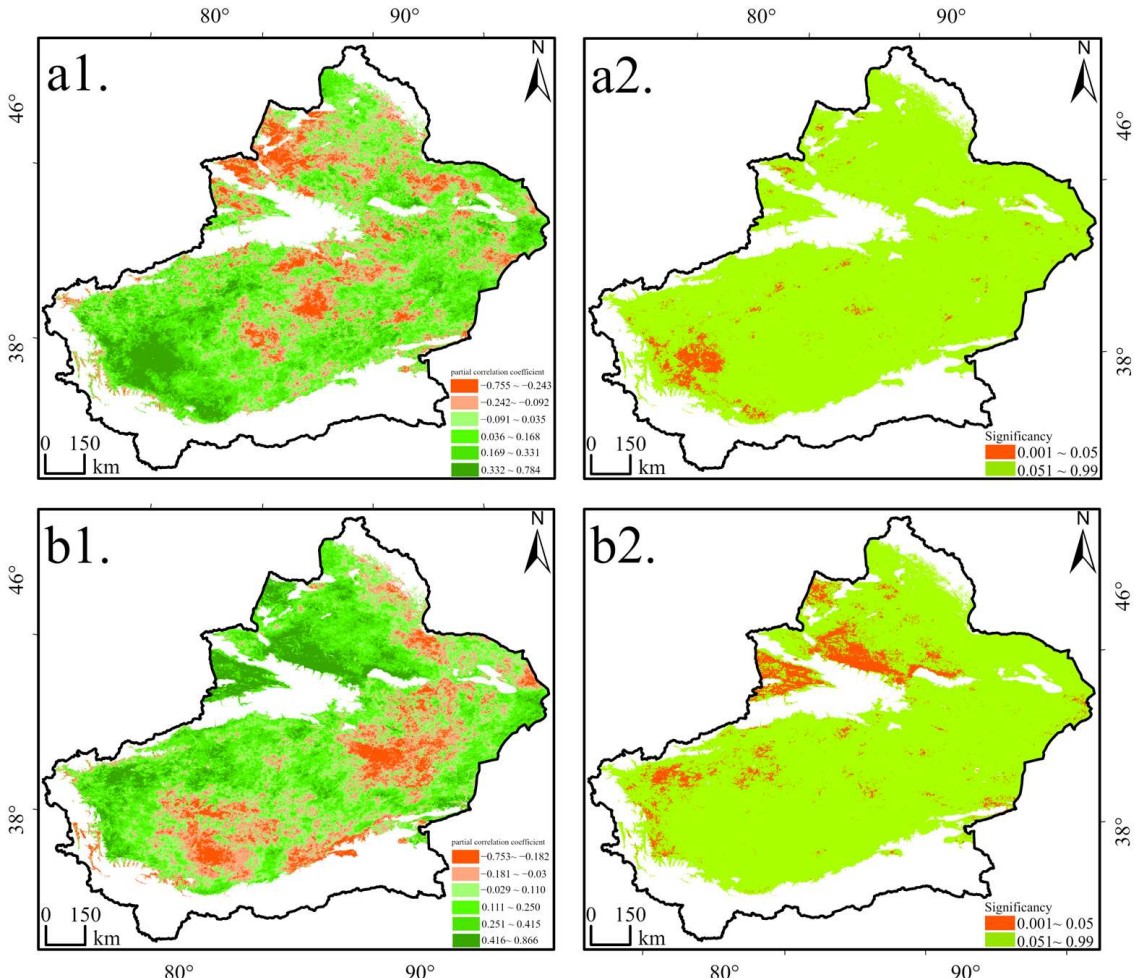

**Figure 14.** Partial correlation analysis results in spring. Note: (**a1**) is the partial correlation results of precipitation, (**a2**) is the significancy results of precipitation, (**b1**) is the partial correlation results of temperature, (**b2**) is the significancy results of temperature.

## 4. Discussion

### 4.1. Spatiotemporal Distribution Characteristics of RSEI

The scientific assessment of regional ecological environment quality and timely grasp of the trend of ecological, environmental change have significant impacts on the long-term and stable development of the region [18,55]. To achieve a harmonious relationship with the ecological environment and maintain the stability of the ecological system, it is necessary to accurately plan economic development, which gives absolute protection to the ecological environment. Previous studies have proved that the RSEI is an efficacious way to evaluate the ecological quality of the region frequently and at a large scale [29,30]. According to previous studies based on the RSEI, the RSEI performances in arid areas are relatively low and unstable, while the RSEI performances of coastal areas have shown high and stable values. Studies have also shown that the overall ecological quality distribution of China presents a spatial pattern of "good in the Southeast and poor in the Northwest" [56]; the areas with good ecological quality conditions are mainly concentrated in the plains and mountainous areas with lower altitudes, superior water and heat conditions, and abundant rainfall [35]. Meanwhile, the areas with poor ecological quality conditions are mainly located in arid and high-altitude inland areas, such as the Inner Mongolia Plateau, Qinghai–Tibet Plateau, Tarim Basin, and Junggar Basin [31,32]. Xinjiang is located in the inner inland of China, with a spatial terrain of "three mountains and two basins". As a typical arid area with the longest distance from the ocean, the extremely low precipitation

in Xinjiang has led to low vegetation coverage, low soil water storage capacity, and high land surface temperature. In this case, the ecological environmental quality in Xinjiang is relatively low and sensitive to human action and environmental changes [57]. According to the terrain spatiality of Xinjiang, the difference in ecological status between Northern Xinjiang and Southern Xinjiang is large, and the precipitation in Northern Xinjiang is higher than in Southern Xinjiang; with this privilege, the vegetation coverage and soil water storage capacity of Northern Xinjiang are better than Southern Xinjiang, and this makes the overall ecological quality status of Northern Xinjiang better than of Southern Xinjiang in the summer [41]. However, in the spring, the situation is the opposite. Because of the high precipitation in Northern Xinjiang, the winter is longer than in Southern Xinjiang. Especially in the spatial places in Altay and Yili, the winter lasts until the end of April. In this case, the large areas of snow and glaciers had an obvious impact on the RSEI results, and there was a large area of low RSEI values in Northern Xinjiang. The high RSEI levels in Altay and Yili were decreased, indicating that the ecological quality in these areas is low in the spring.

The overall evaluation of ecological quality in this study's area is helpful for a better understanding of the change trend of ecological quality. In this case, the from–to change analysis is helpful for vividly visualizing the change trends of the RSEI level in this study's area. The change trend of the RSEI level in this study's area indicates that the changing trend was dominated by a positive changing trend, such as a low RSEI level transforming to a high RSEI level; this situation was obvious in the area around the deserts, such as the Taklamakan Desert, Gurbantunggut Desert, and Kumtag Desert. Although the RSEI performance of Xinjiang was not high, the positive change trend of the RSEI level indicates that in the past 21 years, the overall condition of RSEI in this study's area has improved [48].

### 4.2. Ecological Quality Persistency and Changing Trend in Xinjiang

Changing trend analysis based on long-term data collection is helpful for detecting the subtle change trends of the original data. Based on the overall analysis of the CV, Sen's, M–K test, and the Hurst index, the changing trend persistency and significance of ecological quality in this study's area were found [24,25,48]. First, the results of the CV indicate that the CV value of Xinjiang was concentrated between 0 and 0.02 and distributed in the large part of this study's area, which means that the ecological quality in Xinjiang was relatively stable. Moreover, the higher CV values were distributed in the desert and wilderness in this study, and this result is in accordance with the actual situation of this study's area. According to the results of the significance test, it was found that there was a significant increase in the ecological quality status in Southern Xinjiang in the northwestern part around the Tarim Basin in the last 21 years in summer; in addition, in the spring, the significant increase in ecological quality was mostly concentrated in the southeastern and northern parts of this study's area. According to the persistency of this study's area, it was found that in the last 21 years, the overall ecological quality status of this study's area tended to increase sustainedly both in the summer and spring. The comprehensive analysis of the stability, persistency, and significance test of ecological quality in this study's area showed that in the last 21 years, the ecological quality of Xinjiang was relatively stable both in summer and spring. The area around the Junggar Basin and Kanggashun Gobi showed significant decreasing trends in summer, which means that the ecological quality in these areas tended to be sensitive; in this case, these areas were defined as the warning areas that require spatial protection. In spring, the center of the Tarim Basin, deep in the Taklamakan Desert, was the most unstable area; the area showed a sustained and significant decreasing trend, which was divided into warning areas.

### 4.3. Partial Correlation of Climate Factors between Ecological Quality

The severe natural conditions, special terrain, and climate characteristics make the ecological quality of arid areas extremely fragile and sensitive. In this case, natural factors such as climate and terrain are regarded as the key influencing factors impacting ecological

quality in arid inland by scholars. The special terrain of "three mountains and two basins" with the longest distance from the ocean made the living environment of Xinjiang relatively dry. In this case, the climatic factors turned out to be the most important factor impacting the ecological quality. Based on this situation, this study chose climatic factors such as temperature and precipitation to process the partial correlation analysis between the ecological quality in this study's area. The partial correlation of the climatic factors between ecological quality in summer indicates that the precipitation is positively correlated in most parts of this study's area, and this situation is significantly correlated in the southwestern part, which is mostly desert, wilderness, and the oasis around the Tarim Basin. The temperature in summer positively affects the oasis around the Tarim Basin, and this is significantly correlated. In spring, the precipitation is positively correlated with the ecological quality in most of the region, and this correlation trend is obvious in the oasis around the Tarim Basin. The temperature is positively correlated in the middle and southwestern parts of this study's area, and this correlation trend is obvious in the northwestern region.

### 4.4. Research Limitations and Future Work

Both the accuracy and efficiency of RSEI in the research of ecological quality evaluation have been proved in multiple studies in different regions since its first proposal; in addition, the Landsat data set and MODIS data products were chosen as the original data for RSEI construction. In this study, considering the largeness of the study area, the MODIS data products were selected for RSEI construction instead of the Landsat data set for the reasons listed below. First, in this study, the seasonal RSEI of this study's area in summer and spring needed to be constructed, and quality and quantity were necessary for the selected data. Although the Landsat data sets have higher spatial resolution than the MODIS data products, the data provided by the Landsat satellite are original images with cloud, resulting in low quality and quantity of data that met this study's requirements. However, the MODIS data products have a higher time resolution and provide the integration of corrected data in 8 days, which guarantees data quality. Second, we needed to acquire data from three types of satellites with different sensors for the Landsat data set during this study; this would affect the data continuity and further influence the results of the RSEI. Therefore, the high time resolution and data continuity of the MODIS data products were the main reasons why they were chosen for RSEI construction.

### 5. Conclusions

This study was dedicated to filling the gap in the evaluation of ecological quality in arid areas, especially in large arid areas. Due to the typical arid area characteristics of Xinjiang, the results of this study are able to provide a scientific basis for the environmental management of Xinjiang and the sustainable development of the region. The ecological quality status of Xinjiang was evaluated using the RSEI from 2000 to 2020, and the ecological quality changing trend was processed based on the original RSEI results by the CV, Sen's slope analysis, the M–K test, and the Hurst index, both in spring and summer, in the last two decades. The following conclusions were drawn:

(1) Both in the summer and spring, the high RSEI values of this study's area were concentrated in the oasis around the basin. The low RSEI values were concentrated in the desert and wilderness;

(2) The distribution characteristics of the RSEI levels were dominated by poor and fair values, mostly in the desert and wilderness, while the high RSEI values were concentrated in the oasis around the basin both in the summer and spring;

(3) The ecological quality change trend in Xinjiang was relatively stable and positive. The area that tended to decrease was smaller than the area that tended to increase. The small areas around the Junggar Basin, Turpan Basin, Hami Basin, and Tarim Basin are the areas that tended to significantly decrease;

(4) The precipitation was positively correlated in the northwest part of this study's area in summer and in the eastern part of this study's area in the spring; the temperature

was positively correlated in the northeast part of this study's area both in the summer and spring.

**Author Contributions:** Conceptualization, methodology, formal analysis, visualization, writing—original draft, Y.A.; resources, funding acquisition, conceptualization, supervision, A.K.; conceptualization, investigation, methodology, H.L.; software, investigation, X.Z.; graphic editing, investigation, B.W.; methodology, investigation, Y.Z.; methodology, investigation, M.A. All authors have read and agreed to the published version of the manuscript.

**Funding:** This work was funded by the Third Xinjiang Scientific Expedition Program, grant number (NO. 2021xjkk0905), the National Natural Science Foundation of China, grant number (NO. 42361030), and the Postgraduate Research and Innovation Project of Xinjiang Normal University (XSY202301012).

**Data Availability Statement:** Data are contained within the article.

**Acknowledgments:** The authors are grateful to the reviewers for their constructive comments.

**Conflicts of Interest:** The authors declare no conflict of interest. The funders had no role in the design of this study, in the collection, analyses, or interpretation of data, in the writing of the manuscript, or in the decision to publish the results.

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
