# Peer review of "Evaluation of Ecological Quality Status and Changing Trend in Arid Land Based on the Remote Sensing Ecological Index: A Case Study in Xinjiang, China"

_forests, doi:10.3390/f14091830_

Round 1

Reviewer 1 Report

Dear authors,
The manuscript "Evaluation of ecological quality status and changing trend in arid inland based on the Remote Sensing Ecological Index: a case study in Xinjiang, China” is interesting. In this manuscript, the authors investigate the ecological quality and its changing trend of Xinjiang based on the remote sending ecological index (RSEI) to determine the correlation of climatic factors and ecological environment quality in arid region. They found that ecological quality in Xinjiang is relatively stable with a trend of sustained increase both in summer and spring, and climate affects the ecological quality. The purpose of this study is clear, the data collection is reasonable and feasible, and the sampling point coverage is rich and comprehensive. The analysis method of the article is sound. And the discussion content is in detail and sufficient. However,
the language must be revised by the native English speaker, and there are some issues that to be explained and solved before possible acceptance in the ‘Forests’ journal. Thus, the manuscript needs major revisions

 Comments and suggestions about manuscript

Title

Line 3: Please replace "inland” with “land”.

Abstract

1-      Kick-start the abstract by highlighting the importance of reported work. Briefly discuss the methods used and then discuss key results. Moreover, the whole abstract was hard to read due to the lack of logic and clear clues for their objectives and scientific questions. The abstract does not clearly report the study's main findings; more specific information is required in the abstract instead of general expressions. What is the reason/motivation/research gap? This information was missed in the Abstract.

2-      Line 34-35: Keywords must be rewritten and arranged alphabetically to enhance the visibility of your article.

Introduction

The introduction of the present study only mentions why this study is important and the need for that specific region. What about scientific merit? I did not list all the points, but the introduction needs to be rewritten carefully.

Please rewrite the Introduction considering these questions/points:

1-      What scientific questions have you addressed?

2-      The literature and the data introduced in the Introduction section should finally be compared with the obtained results for discussion.

3-      Scientific community encourages hypothesis-driven research. However, a relevant hypothesis for the study is missing from the introduction. Please add your research hypotheses at the end of the introduction.

4-      Line 73-80: Too long sentence. Please rephrase and split it.

5-      Line 80-86: Another too long sentence. Please rephrase and split it. Please check the whole manuscript and split lengthy sentences for easy understanding of readers.

6-      The authors are encouraged to consult and cite recent literature in the introduction.

https://doi.org/10.3390/f13010100, Forests, https://doi.org/10.1016/j.scitotenv.2023.165394, Science of The Total Environment.

Please merge section 2 and 3 under one heading (Methodology or Materials and Methods) and then rearrange the contents accordingly.

1- After ‘The Xinjiang Uygur Autonomous Region’ (in line 156) the abbreviation (Xinjiang) is needed.

Results

1-      Line 193: The title of Table 1 should be renamed following the title of subsection 2.2.1.

2-      Line 194: The title of subsection 2.2.2 seems incorrect, please check and rewrite.

3-      Line 285: The first line should be indented by two characters, please correct and check the whole manuscript.

4-      Line 456: I don’t think the sentence is finished completely, please check the sentence.

Discussion

The Discussion is fine.

Conclusions

Line 615-634: Please rewrite the precise Conclusions section based on the key findings of your study for the easy understanding of readers.

 Hopefully, these suggestions will help you to improve your Article.

Good luck.

 Moderate editing of English language required

Reviewer 2 Report

The article presents the results of the already traditional studies of changes in the state of the natural environment of the region over a certain period of time. Well-known sources of information on climate and vegetation cover (mainly the RSEI remote sensing index) were used, and generally accepted statistical methods of data analysis were applied. There are no significant comments on the methodological part of the work. The results of the spatial-temporal differentiation of RSEI, its correlation with temperature and precipitation, are interesting for the territory under consideration. At the same time, it cannot be said that the results were unexpected. They completely lie in the logic of the development of natural processes: in desert areas, the indicators are "worse", in oases and river basins – "better". Unfortunately, it is not explained why there were more "good" territories during the study period. Perhaps this is due to changes in precipitation and temperature, so illustrations of the trends of these parameters in the areas where the most active changes occur would be appropriate. The possible reduction of anthropogenic load during this period, which could also have an impact on the increase in the area of "good" territories, is also not discussed. In this regard, the division of territories into "bad" and "good" in ecological terms is not entirely clear. This is an assessment only from the point of view of a person, his needs. In fact, the article objectively shows that for desert territories the RSEI index is lower than for wetter ones (oases, river basins) purely by natural parameters, anthropogenic activity is not considered. Thus, the results of the work reflect the natural features of the functioning of the territory, which in itself is also important. It is desirable that the reasons for the results obtained be considered in more detail in the discussion. I think in general that the work is of scientific and practical interest for this region and leaves a positive impression.

As a small note: the link in line 348 [0,1] is not quite clear.

Reviewer 3 Report

Some parts of the manuscript can be improved. The main point concerns the novelty of the paper. I did not find the explanation what the place of the investigation is in the science. I mean that the novelty should be emphasized in comparison with the results from other studies (as in Introduction, as in Discussion).

I also have some suggestions.

1. The Introduction contains a lot of unnecessary information. I guess it can be shortened. However, please the emphasized the novelty of your research.

2. Line 148. Please also add a few sentences here describing how the rest of the paper is organized. For example:  The rest of the paper is organized as follows. In Section 2, …, Section 3… etc.

3. Figure 2 is quite poor prepared, there are many mistakes here. For example, persistancy not presistancy, negative and positive correlation must be changed with each other, capital and small letters, etc. Check it, please.

4. Figure captions should be written more fully, for example, as for Figures 10,11.

5. What is the importance (scientific and applied) of obtained results in Concludion. It should 1-2 sentance about it.

In general, the paper is poorly written, i.e. it has a lot of mistakes/typos: small letter for headers (Section 5.3), goegraphical objects (for instance,line 621).

The Author have to read the text and check it more carefully. 

Minor editing of the English language is required.

Reviewer 4 Report

The paper is good in general; the sequence of ideas is consistent and well written, the research questions are straightforward, and the research methodology and findings of the researcher are suitable. The references used by the author are good. Also, for the ideas in the article, I will rate this paper 80%. The discussion section provides valuable insights into the implications of the findings. However, it would be helpful to include a comparison of the current study's results with previous research to highlight any similarities. 

Round 2

Reviewer 1 Report

Thanks to the authors for critically addressing my concerns. I am satisfied with the current version. 

Reviewer 3 Report

The manuscript can be published after editorial revision.